# Effect of Tricarboxylic Acids on the Formation of Hydrogels with Melem or Melamine: Morphological, Structural and Rheological Investigations

**DOI:** 10.3390/gels8010051

**Published:** 2022-01-10

**Authors:** Pradip Kumar Sukul, Puspendu Das, Gopal Lal Dhakar, Lalmohan Das, Sudip Malik

**Affiliations:** 1Department of Chemistry, Amity Institute of Applied Sciences, Amity University Kolkata, Action Area-II, Kolkata 700135, India; pksukul@kol.amity.edu; 2School of Applied and Interdisciplinary Sciences, Indian Association for the Cultivation of Science, 2A & 2B Raja S. C. Mullick Road, Kolkata 700032, India; psupd2@iacs.res.in (P.D.); dhakargopal113@gmail.com (G.L.D.); daslalmohan95@gmail.com (L.D.)

**Keywords:** melem, hydrogel network, symmetrical arrangements, viscoelastic properties

## Abstract

Herein, aggregation behaviors of melem or melamine in the presence of three symmetric carboxylic acids (1,3,5-tris(4-carboxyphenyl)benzene (TPCA), 1,3,5-benzene-tri-carboxylic acid (BTA) and 1,3,5-cyclohexane-tri-carboxylic acid (CHTA)) have been performed to check the influence of acid on the formation of aggregated structures which have been investigated by optical microscopy, FESEM, FTIR, XRD and viscoelastic properties have been explored with rheological studies. Interestingly, melem, that has limited solubility in aqueous medium, forms aggregation that leads to the formation of hydrogels with TPCA. More significantly, hydrogel is formed here by matching the size selectivity. Melem forms hydrogel with only large tricarboxylic acid, whereas melamine produces hydrogel with any kind of its counterpart from small to large tricarboxylic acid derivatives. Present investigations and results provide the strategy of design of organic self-assembled materials having two component systems.

## 1. Introduction

Hydrogels are three-dimensional polymeric networks, which are hydrophilic and cross-linked via covalent or noncovalent interactions [1,2,3,4,5]. There is great interest in biomedical applications of hydrogels, as they are homogeneous soft materials with physical properties similar to soft tissues, and they have shown promise in a number of areas including sensors, separation systems, and biomaterials [6,7,8,9]. Recently, significant efforts have been focused on stimuli-responsive hydrogels due to their abilities in response to the external triggers like pH [10], temperature [11], and light [12], making them a class of important organic materials for applications in devices and medicine. However, physical (noncovalent) hydrogels are generally less stable and have poorer mechanical properties compared to covalent hydrogels [13]. It is believed that two-component gel may be the alternative choice to improve the mechanical properties with the judicious introduction of the functional groups onto the individual component. Two components in the hydrogel may interact with each other to form a complex utilizing hydrogen bonding [14,15], electrostatic action [16], donor–acceptor interactions [17] and metal coordination [18], then further aggregate via intercomplex interactions, e.g., hydrogen bonding, π–π interactions or van der Waals forces, to generate the fully three-dimensional networks that eventually produce the formation of semi-solid gel.

Melamine is an excellent building block for constructing well-designed functional supramolecular self-assemblies because of its rigid and planar structure together with multiple hydrogen bonding and π–π stacking ability [19,20,21,22,23]. Melamine produces a very nice hydrogen-bonded rosette motif, the CA-M (CA, Cyanuric Acid and M, Melamine) adduct that is certainly the best studied. Whitesides and co-workers investigated in detail the self-assembly of cyanuric acid and melamine into the respective rosette structures [24]. By employing *s*-triazine molecules with sterically bulky substituents or allowing preorganization of building units by interconnecting them with a hub linker, the formation of rosettes can be favored over the formation of tapes [25,26]. The 1,3,5-triazine ring motif (*s*-triazine, C_3_N_3_) has shown extensive utility in synthetic chemistry [27], coordination chemistry [28,29], and optical and magnetic studies [30,31]. Over the last decade, s-triazine played a key role in molecular routes to generate carbon nitride (CN*_x_*) materials investigated by many groups [32,33,34]. These s-heptazine and cyameluric ring structures were first postulated as a component of the polymer melon, [-C_6_N_7_(NH_2_)-NH-]*_n_*, by Pauling and Sturdivant in 1937 [35]. However, the crystal structure of molecular *s*-heptazine C_6_N_7_H_3_ was first solved by Leonard et al. in 1982 [36]. It has been realized that thermal condensation of melamine is the most convenient route to prepare *s*-heptazine units. The cyameluric nucleus (C_6_N_7_), which is the structural pattern of the heptazine core, is present in melem and melon. Heptazine units are also the building blocks of graphitic g-C_3_N_4_ [37,38,39,40]. The study of carbon nitride-type compounds has become an attractive field, as these units offer several advantages of the higher thermal and chemical stability, catalytic behaviors and semiconducting properties. As a result, scientists or technologists have realized the potential of these units for numerous applications [41,42,43,44]. In some cases, melamine or its derivatives could form supramolecular hydrogels [45] or organogels [46], a class of robust functional soft materials which have received considerable recent attention, integrating one or several imide components due to the ability of formation of triple hydrogen bonding. However, owing to the major problem of solubility, self-assembly study of melem in water is rarely reported [47,48]. Herein, we have rationally picked up two triamines melamine (MM) as well as melem (MEM) and three tricarboxylic acids (TPCA: 1,3,5-tris(4-carboxyphenyl)benzene, BTA: 1,3,5-benzene tricarboxylic acid, and CHTA: 1,3,5-cyclohexane tricarboxylic acid) for the gelation study in which all have C_3_ symmetric functional groups (Figure 1). and we have also explored that not only symmetrical position of complementary interactions but also recognition of site and size are crucial for the creation of one-dimensional self-assembled fibrils that eventually lead to the formation of gelation in aqueous medium.

## 2. Results and Discussion

Melem (s-heptazine, MEM) was synthesized according to Appendix A by the thermal condensation of melamine at 690 K in a closed oven and successfully characterized by solid-state MAS NMR, MALDI-TOF mass, FTIR, and TGA (Appendix A). 1,3,5-Tris(4-carboxyphenyl)benzene (TPCA) was synthesized according to Appendix A and characterized by ^1^H NMR, ^13^C NMR, HRMS, and FTIR [15]. Herein, we have designed and chosen two triamines (MEM and MM) and three tricarboxylic acids (TPCA, BTA, and CHTA) for the gelation study in which all have C_3_ symmetric functional groups (Figure 1). Within these two amines and three acid moieties, the following four sets of bicomponent systems (Table 1) produce hydrogel. At room temperature, melem is not soluble in water; however, at high temperature and in the presence of TPCA, it goes to the solution. As the temperature of the solution lowers to room temperature, it transforms to hydrogel, which is observed by the test-tube inversion method (Figure 1) and it has been later confirmed by rheological studies. Using the similar method, the other three sets of gels have been prepared by taking the following ratio (Table 1) of amine and acid in water.

### 2.1. Morphological Study

The microstructures of the supramolecular hydrogels have been investigated by optical microscopy and FESEM. Most interestingly, the molecular structure and the ratio of the two components have a pronounced effect on the formation of microscopic structure and macroscopic properties of the hydrogel. As seen from Figure 2, all sets of hydrogels produce long fibers which lead to the formation of intermolecular hydrogen–bonded three–dimensional scaffolds [18,19,20,22,49]. Exceptional results have been observed for MM:CHTA 1:1 and it shows one-dimensional crystal–like morphology along with the presence of a gel fibrillar network. Owing to the small size of the crystal, diffraction is not possible. FESEM images show that there are dramatic changes in morphology on tuning the ratio of the two components as well as molecular structure. MEM:TPCA gels in all compositions reveal tubular morphology having a diameter more than 1 µm (Figure 3a,b). MM–TPCA gel gives a fibrous network structure of a width in the range of 100–200 nm (Figure 3c). For a MM-BTA gel system, interesting results have been obtained by varying the ratio of the two components. As observed from Figure 3d, MM:BTA in a molar ratio of 1:1 shows a rod-like structure. When the content of triamine is increased (MM:BTA 3:1), however, long belts having a width of ~5 µm are the main motifs (Figure 3e) produced in the hydrogel. Similarly, if the molar ratio of triacid (BTA) is increased, a belt-like morphology is a dominant feature. For all molar ratios, MM–CHTA gel systems produce a mixture of belt and rhombohedral crystal-like structure having a width in the order of micrometer (Figure 3f). HRTEM images resemble the similar observations for the hydrogel systems (Appendix A).

### 2.2. FTIR Study

To investigate the intermolecular hydrogen bonding during self-assembly, all the dried gel networks have been characterized by FTIR spectroscopy (Figure 4). In pure TPCA, the non-hydrogen-bonded carbonyl-group stretching vibration appears at 1690 cm^−1^, whereas in the dried gel, due to strong intermolecular hydrogen bonding, it appears at 1633 cm^−1^ for MEM:TPCA 1:1, 1618 cm^−1^ for MEM:TPCA 1:3, and 1629 cm^−1^ for MEM:TPCA 3:1. For the MM:TPCA gel system (Figure 4b), the carbonyl stretching vibration in the dried gel appears at 1661 cm^−1^ for MM:TPCA 1:1, 1672 cm^−1^ for MM:TPCA 1:3, and 1647 cm^−1^ for MM:TPCA 3:1. In the case of MM:BTA (Figure 4c), pure BTA shows carbonyl stretching at 1722 cm^−1^. In dried gel, the frequency of carbonyl stretching is shifted to a lower wave number: 1663 cm^−1^ for MM:BTA 1:1, 1670 cm^−1^ for MM:BTA 1:3 system, and 1688 cm^−1^ for MM:BTA 3:1. Pure CHTA reveals the carbonyl stretching vibration at 1695 cm^−1^, whereas the network of dried gel shows carbonyl stretching vibration at 1638 cm^−1^ for MM:CHTA 1:1, 1690 cm^−1^ for MM:CHTA 1:3, and 1672 cm^−1^ for MM:CHTA 3:1 (Figure 4d). Pure MM produces stretching vibration at 1650 cm^−1^ (C=N) and 1548 cm^−1^ (R-NH_2_ bending). In dried gels, these stretching vibrations appear at 1607 cm^−1^ and 1514 cm^−1^. All of the four hydrogel systems have expressed their strong hydrogen-bonded -O-H frequency as a broad stretching vibration above 3300 cm^−1^.

### 2.3. XRD Study

X-ray diffraction studies have frequently been performed for ascertaining the molecular packing of gelator molecules in gel networks. XRD patterns of all the pure compounds in powered state and dried gel networks are provided in Figure 5. XRD patterns of MEM and TPCA are characterized by a group of sharp reflections, indicating that the samples are all crystalline in nature. MEM–TPCA dried gels in all compositions show three main reflection peaks at 2θ = 4.94°, 8.93°, and 10.82°. This result suggests the formation of 1D cylindrical aggregates. XRD patterns of MM:TPCA 1:1 and MM:TPCA 3:1 gel exhibit the diffraction peaks at 2θ = 3.8°, 7.55°, 11.85°, 19.1°, 26.9°, and 30.75° (Figure 5b). Generation of periodic peaks suggests the 1D lamellar structures of the aggregation [22,45], whereas MM:TPCA 1:3 shows several sharp peaks for the crystalline nature of the fibers. XRD pattern of the MM-BTA gel system reveals interesting results, in that the change of molar ratio of individual components has a significant influence on the molecular aggregations. When the molar ratio of the triamine and triacid is 1:1, the many sharp diffraction patterns are seen over the entire region (Figure 5c), exhibiting the 1D rod-like aggregates, which are further matched with the FESEM picture (Figure 3d). Upon variation of the molar ratio, MM:BTA 1:3 or MM:BTA 3:1, the aggregation patterns have changed to layered assembly of belts as it is indicated from the peak ratio of 1:1/2:1/3:1/4. Further, the examination of XRD of the dried network of MM-CHTA (Figure 5d) indicates that d values of the MM:CHTA 1:1 dried gel exhibit four main peaks at 1.54, 0.82, 0.54, and 0.42 nm, respectively, and follows a ratio of 1:1/2:1/3:1/4, suggesting the aggregation of MM and CHTA into a layered structure with interlayer distance of 1.54 nm [14,15]. However, it is also noted that there are some XRD signals that appeared in the trace, and these are the indication of mixed crystal structures of the samples, which are in accordance with the finding of anisotropic growth of the aggregates of MM–CHTA in hydrogels, as revealed in FESEM studies (Figure 3f). The profiles of the other two gel systems are similar to the one described above. The interlayer distance for MM:CHTA 1:3 is 1.26 nm and, for MM:CHTA 3:1 dried gel, is 1.25 nm. The similarity reveals the mixed crystal structure nature of these three samples.

### 2.4. Fluorescence Study

As the value of optical density of gel at any composition is above the instrumental limit of UV-Vis spectrophotometer, fluorescence excitation and emission investigations have been performed for the individual component in water and corresponding network of hydrogel (Figure 6). MEM in water produces the excitation and emission peaks at 370 nm and 431 nm, respectively, whereas TPCA shows the excitation and emission peaks at 348 nm and 407 nm, respectively. However, after making the hydrogel by mixing MEM with TPCA in a 1:1 molar ratio, excitation and emission spectra generate the new peak at 378 nm and 422 nm, respectively (Figure 6a). The red shift of the excitation/emission peak for TPCA occurs after gel formation and it is due to strong π–π stacking between the cores in the gel [15]. From Figure 6b, MM shows excitation and emission peaks at 266 and 370 nm. Upon the formation of hydrogel with TPCA, the excitation and emission peaks appear at 348 nm and 433 nm, respectively. Therefore, the emission band for TPCA in MM–TPCA hydrogel shows a red shift of 26 nm. BTA itself shows excitation and emission bands at 336 nm and 391 nm (Figure 6c). In MM–BTA gel, excitation and the emission band appear at 267 nm and 378 nm, respectively. Surprisingly, BTA reveals blue shifting of the emission band by 13 nm. Pure CHTA in Figure 6d gives excitation and the emission band at 290 and 384 nm, whereas in the MM–CHTA gel, the emission band appears at 363 nm, indicating the blue shifting of the emission peak by 21 nm. It is interesting to note that MM–TPCA gel gives red shifting and BTA−CHTA gel shows blue shifting of the emission peak in the emission spectra. Gel networks composed of microscopic fibers show fluorescent behavior as it is revealed from fluorescence microscopy images of the dried gels upon excitation at 380 nm (Appendix A).

### 2.5. Rheological Studies

Viscoelastic properties of hydrogel networks having different molar ratios have been studied by placing a piece of hydrogel between two plates through deformation-controlled (or stress-controlled) oscillatory experiment in which a periodic deformation ($\gamma \left(t\right)=\gamma_{\circ }cos\left\{\omega t\right\}$, where ω is the angular frequency at time “*t*” and *γ*_ο_ is the deformation amplitude) is induced to have a periodic stress ($\tau \left(t\right)=\tau_{\circ }cos\left\{\omega t+\delta \right\}$, where *τ*_ο_ is the stress amplitude) with the difference of phase of “*δ*” relative to the initial stress (*δ* = 0 for elastic materials, *δ* = 90° for viscous materials and 0 < *δ* < 90° for viscoelastic materials such as semi-solid gel [49,50,51,52].

In the linear region, the preservation of network structure remains intact and these expressions hold good agreement with the experimental results. However, in the nonlinear region where the depletion of network structures may occur, complex values of deformations as well as stress have to consider with the respective imaginary component. The above expressions will be modified to [50,52,53].

Complex deformation:(1)$$\gamma *(t)=\gamma_{\circ }\left(\cos\left\{\omega t\right\}+i\sin\left\{\omega t\right\}\right)=\gamma_{\circ }e^{i\omega t}$$

Complex stress: (2)$$\tau *(t)=\tau_{\circ }\left(\cos\left\{\omega t+\delta \right\}+i\sin\left\{\omega t+\delta \right\}\right)=\tau_{\circ }e^{i\left(\omega t+\delta \right)}$$

Complex modulus of rigidity:(3)$$G*=\frac{\tau *(t)}{\gamma *\;(t)}=\frac{\tau_{\circ }}{\gamma_{\circ }}e^{i\delta }=G^{\prime }+i\cdot G^{\prime \prime }$$
where *G*′ is the storage modulus, i.e., the degree of elastic energy that is reversibly stored upon lifting forces on the hydrogel network and *G*″ is the loss modulus that is dissipated in an irreversible manner due to the viscous character of hydrogel.
$$G^{\prime }=\frac{\tau_{\circ }}{\gamma_{\circ }}\cos\delta \;G^{\prime \prime }=\frac{\tau_{\circ }}{\gamma_{\circ }}\sin\delta$$

The ratio of loss modulus (*G*″) to storage modulus is familiarly known as dissipation factor (tan *δ*). If the value of “tan δ” is greater than unity, it reveals the viscous behavior of systems, and if the value of “tan *δ*” is less than unity, it indicates the elastic nature of a hydrogel network. Influences of different morphologies on the viscoelastic properties (Figure 3) of a hydrogel network have been characterized by oscillatory rheology at their corresponding minimum gelator concentrations. The frequency sweep has been measured by applying the constant oscillating strain at 1% (Figure 7). The storage modulus *G*′ and the loss modulus *G*″ curves of all the hydrogels demonstrate that values of *G*′ are larger than values of *G*″ over the entire angular frequency range. These results indicate that all hydrogels possess the network structures that behave as a semi-solid-like material [54]. From the magnitude of *G*′, it can be revealed that the hydrogel network of MEM–TPCA is mechanically robust in nature. Such a high mechanical property originated from small molecular network is not generally observed [55]. Plots of both moduli vs. % strain plots for the hydrogel network provide the gradual transformation of semi-solid network to viscous liquid. The strain at the transformation (denoted by arrow in Figure 8) is recognized as the limit of linearity (*γ_0_*) or critical strain of gel. Furthermore, utilizing absolute values of *G′* and *G″* (Appendix A), the elasticity and the stiffness of each hydrogel network have been estimated (Figure 9). The hydrogel network made of MEM–TPCA (1:1) revealed the highest value of elasticity (1.09 × 10^5^ Pa) among twelve systems, and such a high value of elasticity is extremely rare for a small molecular hydrogel network. Values of stiffness are relatively higher for MEM–TPCA and MM–TPCA systems (Figure 9b) and more rigorous investigations are required to find out the reason of higher values of elasticity/stiffness for these hydrogel networks.

The above results show that melem forms hydrogel only with TPCA, whereas melamine forms hydrogel with all three tricarboxylic acids without any dependency on size. This finding is important in the field of designing molecules for proper self-assembled systems. Figure 10 represents the hydrogen-bonding interaction motifs between the amine and acid molecules. Melamine has the only size matching with BTA. It has no matching with CHTA or TPCA. Melamine has three hydrogen bonding sites with carboxylic acids. That is why three triacid molecules can come from symmetric sides and construct a one-dimensional regular structure. Finally, three-dimensional networks are formed due to aggregation in water medium and gelation takes place. On the other hand, melem has almost a similar size with TPCA. However, it is large in size compared to the size of BTA or CHTA. Melem bears six recognition sites and is large in size; smaller-size BTA can come from six sites and it disturbs the symmetrical arrangements of a network and also enhances critical mass, which leads to precipitation. In spite of having six recognition sites, only three TPCA can come closer to melem in a symmetrical manner to form a one-dimensional regular structure. Steric hindrance is the key factor for only three TPCA to come towards melem, which will be energetically favorable. This phenomenon is further supported by the fluorescence study, which suggests red shifting in case of MEM–TPCA gel due to large slippage between pi-conjugated benzene cores.

## 3. Conclusions

Aggregations and gelation ability of different triamines (melem and melamine) and tricarboxylic acid derivatives have been investigated in the present manuscript. Melem, in spite of limited solubility in water, has aggregated as well as formed hydrogel owing to the influence of its counterpart TPCA tricarboxylic acid. More interestingly, hydrogel is formed by matching the size selectivity. Melem forms hydrogel with only large tricarboxylic acid, whereas melamine is able to form hydrogel individually with all acidic counterparts ranging from small to large tricarboxylic acid derivatives. The formation of hydrogel network has been investigated by optical microscopy, FESEM, FTIR, XRD, and viscoelastic properties have been explored with rheological studies. Among all systems, optimum elasticity is obtained for the MEM–TPCA system, and such a high value of elasticity obtained from small molecular system is extremely rare. This present study will provide the strategy of design of self-assembled materials made of small organic molecules. Like polymeric melon-type carbon nitrides (CNs), melem hydrogel matrix could be an efficient photocatalytic system.

## 4. Materials and Methods

### 4.1. Materials

Melamine (MM), 4–Bromoacetophenone, potassium pyrosulphate (K_2_S_2_O_7_)_,_ ethanol (EtOH), Tetrahydrofuran (THF) n-butyl lithium (n−BuLi) and all other chemicals and solvents were purchased from Merck. Gelling water was HPLC grade and purchased from Spectrum. Melem was synthesized following the reported procedure as in Appendix A. All synthesis procedures of tricarboxylic acids were carried out according to Appendix A. All the compounds were fully characterized by MALDI−TOF mass spectrometry, ^1^H NMR spectroscopy (300 MHz), ^13^C NMR spectroscopy (75 MHz), solid-state NMR spectroscopy, FTIR and TGA.

### 4.2. Synthesis

#### 4.2.1. Synthesis of Melem (MEM)

MM was prepared in the laboratory according to the previous reports [39,56]. MM was initially weighted in porcelain and it was put into an oven at 390 °C for 2 days. Product was collected from the porcelain carefully. Yield: 35%.

Mass (MALDI-ToF): calcd 218.12 found 219.20 (M + H)^+^.

#### 4.2.2. Synthesis of 1,3,5-Tri(4-bromophenyl)benzene (3)

4-Bromoacetophenone (10 g, 50.25 mmol), 0.5 mL of H_2_SO_4_(c) and K_2_S_2_O_7_ (15 g, 59 mmol) were heated at 180 °C for 14 h under nitrogen atmosphere. The resulting crude solid was cooled to room temperature and refluxed in 50 mL of dry EtOH for 1 h and then cooled to room temperature. The solution was filtered and the resulting solid was refluxed in 50 mL of H_2_O, giving a pale-yellow solid that was filtered. The crude product was dried under vacuum and recrystallized in CHCl_3_. Yield: 70%. ^1^H NMR (300 MHz, CDCl_3_): *δ* = 7.53 (d, 6H), 7.60 (d, 6H), 7.68 (s, 3H) ppm. ^13^C NMR (75 MHz, CDCl_3_): *δ* = 122.2, 125.1, 129, 132.2, 139.7, 141.6 ppm.

#### 4.2.3. Synthesis of 1,3,5-Tris(4-carboxyphenyl)benzene (TPCA)

1,3,5-tri(4-bromophenyl)-benzene (3 g, 5.52 mmol) was dissolved in 40 mL of anhydrous THF under N_2_ atmosphere. The stirred solution was cooled to −60 °C, and a 1.6 M solution of n-BuLi in hexanes (10.5 mL, 16.3 mmol) was added dropwise. A red-to-light-green precipitation of the aryl lithium derivative was formed. Freshly prepared and dried gaseous carbon dioxide was passed into the mixture at −60 °C to give a colorless precipitate of the lithium salt. The mixture was allowed to warm and was quenched with 50% aqueous acetic acid. The solid product was filtered and recrystallized from acetic acid to give 0.8 g (33%) of white microcrystals [15]. ^1^H NMR (300 MHz, DMSO−d_6_): *δ* = 8.038–8.045 (d, 12 H), 8.07 (s, 3 H) ppm, ^13^C NMR (75 MHz, DMSO−d_6_): *δ* = 125.5, 127.4, 129.9, 130.0, 140.8, 143.7, 167.2 ppm. Mass (MALDI–TOF) calcd 438.1 found 439.2 [M + H]^+^. FTIR: 3071, 2985, 1691, 1608, 1419, 1318, 1294, 1245 cm^−1^.

### 4.3. Preparation of Hydrogel

The required amounts of melem (MEM), melamine (MM), acid moieties and water were taken inside the hermitically sealed tube, boiled for some time to make a homogeneous solution. At room temperature, melem was not soluble in water, even at high temperature; it was dissolved to make the solution in the presence of acid. As the temperature of solution was lowered to room temperature (25 °C), it formed hydrogel (for melamine only: on lowering the temperature of the solution, it was precipitated out). Using the similar method, the other three sets of hydrogels were prepared with a molar ratio of 1:1 by taking the following weight percentage (Table 1) of amine and acid. Prepared hydrogels were used directly for microscopy and rheological studies. Otherwise, hydrogels used for XRD, FTIR, FESEM, etc., were dried under the pool of air and finally kept in vacuum for 3 consecutive days prior to the experiment.

### 4.4. Methods

#### 4.4.1. Microscopy

The morphology of the network of dried gel was observed through an optical microscope (Leitz, Biomed) under perfectly crossed polarizer and taking the picture through a digital camera (Leica D-LUX 3). FESEM experiments were performed by placing a small portion of network of gel on a microscope cover glass. Samples were dried first in air and then in vacuum and coated with platinum prior to observation. After that, micrographs were recorded by using a Jeol Scanning Microscope JSM-6700F.

#### 4.4.2. Fluorescence Microscopy and Spectroscopy

Gel-phase material was kept on a glass slide, on another cover slip it was placed to thin film, and examined under a fluorescence microscope (OLYMPUS BX-61) in 40× magnification. Fluorescence spectral studies of all the gel samples prepared in a sealed cuvette were carried out in a Horiba Jobin Yvon Fluoromax 3 instrument. The gel samples were directly prepared in a quartz cell of 1 cm path length. Fluorescence excitation acquisition and emission acquisition were taken in slit width of 2/2 nm and scan rate was 0.2 s.

#### 4.4.3. XRD Study

The WAXS studies of the dried gels were performed by a Seifert X-ray diffractometer (C–3000) using nickel-filtered Cu-Kα radiation with a parallel beam optics attachment. The instrument was operated at a 35 kV voltage and a 30 mA current and was calibrated with standard silicon sample. The samples were scanned from 2θ = 2° at the step-scan mode (step size 0.03°, preset time 2 s), and the diffraction pattern was recorded using a scintillation counter detector.

#### 4.4.4. Rheological Study

Rheological experiments were performed with an AR 2000 advanced rheometer (TA Instruments) using cone-plate geometry in a Peltier plate. The plate diameter was 40 mm, with a cone angle of 2 degrees. Two types of experiments were performed: (i) by frequency sweep and (ii) by strain sweep methods. The frequency sweep experiments were made at 25 °C at constant 1% strain. The strain sweep experiment was performed at 25 °C at a constant frequency of 2 rads^−1^. As prepared, hydrogel was used for rheological study and the piece of hydrogel was kept between two plates at 25 °C.

#### 4.4.5. FTIR Spectroscopy

FT-IR spectra of MM, MEM, TPCA, BTA, CHTA and dried gels were recorded using KBr pellets of samples in an FTIR-8400S instrument (Shimadzu). All samples were diluted with KBr and pelletized prior to the experiment. Two independent experiments were performed for each sample.

## Data Availability

The data presented in this study are available on request from the corresponding author.

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
