# Peer review of "Effect of Tricarboxylic Acids on the Formation of Hydrogels with Melem or Melamine: Morphological, Structural and Rheological Investigations"

_gels, 2022, doi:10.3390/gels8010051_

Round 1

Reviewer 1 Report

English language needs improvement, a lot of unclear sentences and typos mistakes, such as “symmemetric”, “maicroscope”. Therefore, for help with English language usage and quality, I strongly recommend that the authors should consult a native English speaker.

Figure 2: scalling bar : micrographs g,h,i

Sentence not clear “After formation gel, in dried gel…”. Please reword.

Figure 4, low resolution, hard to read the wavelength values labeled on the spectra.

Figure 5, low resolution, hard to identify the values included on the XRD diffractograms.

An overview on the potential application of these hydrogels should be included in the conclusion part.

More details about preparation of hydrogels should be provided in section 4.2. The amount (g, moles) of each component should be included.

Which method was used for gel drying? This information is missing from experimental part.

For rheological study is not clear which type of gels were used, dried or swollen state gels.

What about their swelling degree and water vapor sorption? Did the authors investigate their equilibrium swelling behavior?

Author Response

Thank you very much for your valuable comments on our manuscript. We have carefully read your comments and answered to your queries in a one-by-one manner.

Comment1: English language needs improvement, a lot of unclear sentences and typos mistakes, such as “symmemetric”, “maicroscope”. Therefore, for help with English language usage and quality, I strongly recommend that the authors should consult a native English speaker.

Response: We are extremely sorry for the unintentional mistake of typos which are corrected with care in the revised manuscript.

Comment 2 Figure 2: scalling bar : micrographs g,h,i

Response: Scalling bars have been inserted in the Figure2.

Comment 3: Sentence not clear “After formation gel, in dried gel…”. Please reword.

Response: It has been taken care.

Comment 4: Figure 4, low resolution, hard to read the wavelength values labeled on the spectra.

Response: It has been improved.

Comment 5: Figure 5, low resolution, hard to identify the values included on the XRD diffractograms.

Response: It has been improved

Comment 6: An overview on the potential application of these hydrogels should be included in the conclusion part.

Response: It has been included.

Comment 7: More details about preparation of hydrogels should be provided in section 4.2. The amount (g, moles) of each component should be included.

Response: It is mentioned in table 1.

Comment 8: Which method was used for gel drying? This information is missing from experimental part.

Response: It is mentioned in each section of the morphological investigations.

Comment 9: For rheological study is not clear which type of gels were used, dried or swollen state gels.

Response: As prepared hydrogel was used for rheological study and the piece of hydrogel were kept between two plates at 25 °C.

Comment 10: What about their swelling degree and water vapor sorption? Did the authors investigate their equilibrium swelling behavior?

Response: Swollen study is not investigated. Swollen study is generally carried out for cross-linked gel system (chemical gel). For small molecular gelator, we never think of this study as these gels are physical in nature.

Reviewer 2 Report

This manuscript describes the self-assemble behaviour of melem and melamine in presence of 1,3,5-tris(4-carboxyphenyl)benzene, 1,3,5-benzene-tri-carboxylic acid and 1,3,5-cyclohexane-tri-carboxylic acid. My major concern regarding this manuscript is that the authors failed to point the importance of the results obtained.  Moreover this manuscript is part of collection entitled “Recent Advances and Future Perspectives in Organogels and Organogelators” and the compounds gave hydrogels and the manuscript is about hydrogels and hydrogelators. The introduction must be re-written. The results presented in the introduction section should go to the results section. The conclusions should emphasize the importance of this study and of the results obtained.  

Author Response

Thank you very much for your valuable comments on our manuscript and your positive recommendation towards its publication. We have revised the manuscript and tried to improve the text according to your valuable suggestions.

Reviewer 3 Report

The paper reported by Malik et al. describes aggregation behaviors of melem or melamine in presence of three symmemetric carboxylic acids followed by characterization with  the help of FTIR, PXRD, FESEM, etc. The paper is recommended for publication after a major revision.

  1. Abstract: The authors are requested to correct the spelling of microscope.
  2. Introduction: The authors are requested to take care of formatting Vander Waal’s forces. Vander Waal’s forces to be changed to Van der Waals force. I believe the authors will take care of such serious mistake for their future publications.
  3. Table-1: ml to be changed to mL.
  4. The authors are requested to add G’, G’’ value as well as gel melting temperature data from SI in the table.
  5. Line 161: Table 1; please make it bold to maintain the uniformity.
  6. The authors are requested to add TEM and AFM data for the corresponding FESEM images.
  7. Figure 4 and 5: The quality of the graph is very poor. The authors are strongly requested to add high-resolution plot for clear understanding.
  8. The authors are strongly requested to add NMR titration data.
  9. Figure 6: The authors are requested to present in a nicer way. It is too much work for eye. Please add simple plot.
  10. Figure 7 & 8: Figure quality is poor. Please enhance the quality of the figures.
  11. Line 517: Figure 3 will be bold.
  12. MALDI-ToF to be changed to MALDI-TOF. Please correct it throughout the manuscript.
  13. Line 608 & 625: 13C NMR spectroscopy (300 MHz): It can’t be 300 MHz. Please correct it.
  14. Line 614, 630, 633, 646 & 682: Please add space between number and °C.
  15. 13C NMR (DMSO-d6): Please add MHz.
  16. Reference 1: Please use appropriate format.
  17. The authors are requested to add 1H, 13C, mass for each and every synthesized compound.

Author Response

The file is attached for you.

Round 2

Reviewer 2 Report

The authors should  re-written the introduction. The results presented in the introduction section should go to the results section. The conclusions should emphasize the importance of this study and of the results obtained.   

Author Response

Comment_R2: The authors should re-written the introduction. The results presented in the introduction section should go to the results section. The conclusions should emphasize the importance of this study and of the results obtained.

Response: We have checked the section of introduction and have tried to improve it. Importance of the present study has been mentioned already as photocatalyst as we are currently working on its applications.

Reviewer 3 Report

The authors have addressed all the concerns properly. I am happy to see the paper published now. It would be highly appreciated if the authors put TEM images in ESI. 

Author Response

Comment_R3: The authors have addressed all the concerns properly. I am happy to see the paper published now. It would be highly appreciated if the authors put TEM images in ESI.

Response: Thank you for the support. As per your valuable suggestion, HRTEM images are included in the section of ESI.

Round 3

Reviewer 2 Report

The manuscript can be accepted for publication in Gels.